# Cause of Death in Heart Failure Based on Etiology: Long-Term Cohort Study of All-Cause and Cardiovascular Mortality

**DOI:** 10.3390/jcm11030784

**Published:** 2022-01-31

**Authors:** Giosafat Spitaleri, Elisabet Zamora, German Cediel, Pau Codina, Evelyn Santiago-Vacas, Mar Domingo, Josep Lupón, Javier Santesmases, Crisanto Diez-Quevedo, Maria Isabel Troya, Maria Boldo, Salvador Altimir, Núria Alonso, Beatriz González, Antoni Bayes-Genis

**Affiliations:** 1Heart Failure Clinic and Cardiology Service, University Hospital Germans Trias i Pujol, 08916 Badalona, Spain; giosafat.spitaleri@yahoo.it (G.S.); e.zamora@telefonica.net (E.Z.); gecediel@yahoo.com (G.C.); pau.codi@gmail.com (P.C.); nyleve.sv@hotmail.com (E.S.-V.); madote@gmail.com (M.D.); jlupon.germanstrias@gencat.cat (J.L.); jsantesmases.germanstrias@gencat.cat (J.S.); cdiezquevedo.germanstrias@gencat.cat (C.D.-Q.); m.troyasaborido@gmail.com (M.I.T.); boldomaria@gmail.com (M.B.); saltimir.germanstrias@gencat.cat (S.A.); nalonso32416@yahoo.es (N.A.); bgonzalez.germanstrias@gmail.com (B.G.); 2Department of Medicine, Universitat Autonoma de Barcelona, 08193 Barcelona, Spain; 3CIBERCV, Instituto de Salud Carlos III, 28029 Madrid, Spain

**Keywords:** heart failure with reduced ejection fraction, heart failure with preserved ejection fraction, heart failure etiology

## Abstract

We assessed differences in long-term all-cause and cardiovascular (CV) mortality in heart failure (HF) outpatients based on the etiology of HF. Consecutive patients admitted to the HF Clinic from August 2001 to September 2019 (*N* = 2587) were considered for inclusion. HF etiology was divided into ischemic heart disease (IHD), dilated cardiomyopathy (DCM), hypertensive heart disease, alcoholic cardiomyopathy, drug-induced cardiomyopathy (DICM), valvular heart disease, and hypertrophic cardiomyopathy. All-cause death and CV death were the primary end points. Among 2387 patients included in the analysis (mean age 66.5 ± 12.5 years, 71.3% men), 1317 deaths were recorded (731 from CV cause) over a maximum follow-up of 18 years (median 4.1 years, interquartile range (IQR) 2–7.8). Considering IHD as the reference, only DCM had a lower risk of all-cause death (adjusted hazard ratio (aHR) 0.68, 95% confidence interval (CI) 0.56–0.83, *p* < 0.001), and only DICM had a higher risk of all-cause death (aHR 1.47, 95% CI 1.02–2.11, *p* = 0.04). However, almost all etiologies had a significantly lower risk of CV death than IHD. Among the studied HF etiologies, DCM and DICM have the lowest and highest risk of all-cause death, respectively, whereas IHD has the highest adjusted risk of CV death.

## 1. Introduction

Despite improvements in prognosis over the last three decades, patients with chronic heart failure (HF) remain at high risk of death [1,2,3]. Multiple factors, such as left ventricular ejection fraction (LVEF) [4], influence survival, but evidence on the prognostic impact of HF etiology is scarce.

Current evidence suggests that some etiologies are associated with higher mortality rates than others [5,6,7,8,9]. In particular, ischemic heart disease carries a significantly poorer prognosis than other HF etiologies. However, these studies were conducted more than 10 years ago and had a limited follow-up. In addition, they analyzed only the occurrence of all-cause mortality without providing further insights on cardiovascular (CV) mortality. As therapies that have been proven to improve survival in HF reduce the incidence of death from CV causes, particularly sudden death and progressive HF, with no benefit on non-CV mortality, it may be useful to identify the etiologies associated with higher risk of CV death in order to select patients that can actually benefit from disease-modifying treatment.

Thus, the aim of the present study was to assess differences in long-term all-cause and CV mortality in a cohort of HF outpatients based on the etiology of HF.

## 2. Materials and Methods

### 2.1. Study Population and Outcomes

All consecutive ambulatory patients admitted to a structured multidisciplinary HF clinic at a university hospital between August 2001 and September 2019 were considered for the study. During the 18-year study period, the clinical pathways and referral geographic area, covering ∼850,000 inhabitants in the northern Barcelona Metro Area, remained stable. Most patients were referred to the HF clinic by the Cardiology or Internal Medicine Departments, and to a lesser extent by the Emergency Department or other hospital departments. The criteria for referral to the HF clinic were HF according to the European Society of Cardiology (ESC) guidelines, regardless of etiology, at least one HF hospitalization, and/or reduced LVEF [10,11]. All patients were seen regularly for follow-up visits at the HF clinic according to their clinical needs and treated according to a unified protocol. Follow-up visits included a minimum of one visit with a nurse every 3 months and one visit with a physician (cardiologist, internist, or family physician) every 6 months. Optional visits were performed by specialists in geriatrics, psychiatry, and rehabilitation [10,11], with the addition of a nephrologist and endocrinologist in recent years. Modes of death were classified as HF progression (worsening HF or treatment-resistant HF in the absence of another cause), sudden death (any unexpected death, witnessed or not, of a previously stable patient with no evidence of worsening HF or any other known cause of death), acute myocardial infarction, stroke, procedural (post-diagnostic or post-therapeutic), other CV causes (e.g., rupture of an aneurysm, peripheral ischemia, or aortic dissection), or non-CV [12]. Fatal events were identified from the patients’ health records from hospital wards, the emergency room, and general practitioners or by contacting their relatives. Data were verified using the databases of the Catalan and Spanish Health Systems, as well as the Spanish National Death Registry (INDEF). Adjudication of events was performed by an ad-hoc committee (JL, MdeA, BG, and MD; PM and GC resolved the possible discrepancies). During the baseline visit, patients provided written consent for the use of their clinical data for research purposes. The study was performed in compliance with the law protecting personal data in accordance with the international guidelines on clinical investigations from the World Medical Association’s Declaration of Helsinki. The local ethics committee approved the study.

### 2.2. Etiology of HF

Eight major groups were identified according to HF etiology: ischemic heart disease, dilated cardiomyopathy (CM), hypertensive heart disease, alcoholic CM, drug-induced CM, valvular heart disease, hypertrophic CM, and other/unknown. For the purpose of this analysis, the latter group was excluded due to the heterogeneity and the limited number of patients in each subtype. Coronary angiography was not performed on a routine basis, but information about the coronary anatomy, if available, was used for the classification.

#### 2.2.1. Ischemic Heart Disease

This group included all patients with a history of prior myocardial infarction or coronary intervention, either coronary artery bypass graft surgery or percutaneous coronary intervention. This group also included patients with a history of chest pain who had pathological Q waves on the electrocardiogram and/or dyskinetic areas on the echocardiogram.

#### 2.2.2. Dilated Cardiomyopathy

This group comprised all patients with depressed LVEF and/or dilatation of the left ventricle when another distinct etiology had not been found despite routine workup, which would have included evaluation for the presence of ischemic heart disease. Myocardial non-compaction was included in this group.

#### 2.2.3. Hypertensive Heart Disease

This group included all patients with arterial hypertension as the only explanation for HF.

#### 2.2.4. Alcoholic Cardiomyopathy

This group included all patients with depressed LVEF and/or dilatation of the left ventricle and a known history of alcohol abuse when another distinct etiology had not been found despite routine workup.

#### 2.2.5. Drug-Induced Cardiomyopathy

This group mainly included patients that underwent chemotherapy at any time and presented with a ≥5% decrease in the LVEF in the presence of symptoms of HF or an asymptomatic decrease in LVEF ≥ 10% but <55% [13].

#### 2.2.6. Valvular Heart Disease

This group included all patients with severe primary aortic or mitral valve disease on echocardiography. Patients with functional mitral regurgitation were classified as ischemic or dilated CM depending on the primary cause of the regurgitation, regardless of the degree of regurgitation. Other patients in this group were patients operated on for mitral or aortic valve disease with signs of left ventricular dysfunction, evidence of pulmonary hypertension, or signs of congestion. Ischemic heart disease was excluded based on history or coronary angiography if coronary artery disease was considered to not be significant enough to be the cause of HF.

#### 2.2.7. Hypertrophic Cardiomyopathy

Hypertrophic CM was defined as a wall thickness ≥ 15 mm in one or more left ventricular myocardial segments as measured by cardiac magnetic resonance or echocardiography and that was not explained solely by loading conditions [14].

#### 2.2.8. Other Etiologies

Other etiologies, such as peripartum CM, tachycardia-induced CM, constrictive pericarditis, and amyloidosis, were not included in the analysis due to heterogeneity, different pathophysiological mechanisms, and limited number of patients in each subtype.

### 2.3. Statistical Analysis

Categorical variables were expressed as absolute numbers and percentages. Continuous variables were expressed as the mean ± standard deviation (SD) or median (interquartile range (IQR)) according to normal or non-normal distributions. Normal distribution was assessed by normal Q to Q plots. Between-group comparisons were made using chi-squared and Fisher’s exact test for categorical variables, and the one-way analysis of variance (ANOVA) or Kruskal–Wallis test for continuous variables, as appropriate. Multivariable Cox regression analysis for all-cause and CV mortality was performed using Fine and Gray competing risk methodology. The HF group with ischemic heart disease was chosen as the reference group. Mortality data from the other HF etiology groups were then compared to the reference group in multivariable analyses. To avoid over-fitting, only age, sex, and variables with *p* < 0.10 in the univariate analysis for each etiology (among HF duration, New York Heart Association (NYHA) class III-IV, diabetes, anemia, renal insufficiency, chronic obstructive pulmonary disease, and peripheral arteriopathy) were included in the multivariable model. Cumulative incidence curves for all-cause and CV mortality were also plotted using a competing risk method. Statistical analyses were performed in SPSS 24 (SPSS Inc., Chicago, IL, USA) and Stata (version 13.0, Stata Corp., College Station, TX, USA). A two-sided *p* < 0.05 was considered significant.

## 3. Results

### 3.1. Baseline Characteristics

Of 2387 patients admitted to our HF clinic from August 2001 to September 2019, 1222 (51.2%) presented with ischemic heart disease, 415 (17.4%) with dilated CM, 231 (9.7%) with hypertensive heart disease, 120 (5.0%) with alcoholic CM, 73 (3.1%) with drug-induced CM, 237 (9.9%) with valvular heart disease, and 89 (3.7%) with hypertrophic CM. The baseline demographic and clinical characteristics of patients according to HF etiology are shown in Table 1. There was substantial heterogeneity among the groups with regard to age, sex, CV risk factors, and HF symptoms. The mean age of the study cohort was 67 ± 13 years and 71.3% of patients were men. Patients with alcoholic CM, drug-induced CM, and hypertrophic CM were significantly older than those with ischemic heart disease, whereas patients with hypertensive heart disease and valvular heart disease were significantly older. In addition, patients with alcoholic CM and those with hypertrophic CM reported less severe NYHA functional class symptoms, whereas patients with hypertensive heart disease and valvular heart disease had a higher NYHA functional class than those with ischemic heart disease.

### 3.2. Association between HF Etiology and Mortality

During a median follow-up of 4.1 years (IQR 2 to 7.8 years) (5.3 years (IQR 2.6 to 9.7 years) for survivors, maximum follow-up of 18 years), 1317 deaths (55.2%) were recorded, including 731 (55.5%) from a CV cause. Table 2 shows unadjusted and adjusted hazard ratios for each HF etiology compared to ischemic heart disease. In the unadjusted analysis, patients with dilated CM (hazard ratio (HR) 0.48, 95% confidence interval (CI) 0.40–0.57, *p* < 0.001), alcoholic CM (HR 0.49, 95% CI 0.37–0.65, *p* < 0.001), and hypertrophic CM (HR 0.27, 95% CI 0.14–0.50, *p* < 0.001) had better survival than those with ischemic heart disease. In contrast, patients with valvular heart disease had the worst survival (HR 1.20, 95% CI 1.01–1.42, *p* = 0.04). After adjusting for potential confounders, survival was better only in patients with dilated CM (adjusted HR 0.68, 95% CI 0.56–0.83, *p* < 0.001) and only drug-induced CM was associated with a higher risk of death compared to ischemic heart disease (adjusted HR 1.47, 95% CI 1.02–2.11, *p* < 0.04). Figure 1 shows survival curves for all-cause death (Figure 1A) and cumulative incidence curves for CV mortality (Figure 1B).

With respect to CV mortality, both the unadjusted and adjusted risk of CV death was lower in patients with dilated (adjusted HR 0.54, 95% CI 0.42–0.70, *p* < 0.001), alcoholic (adjusted HR 0.59, 95% CI 0.37–0.94, *p* = 0.03), drug-induced (adjusted HR 0.42, 95% CI 0.22–0.83, *p* = 0.01), or hypertrophic CM (adjusted HR 0.40, 95% CI 0.18–0.90, *p* = 0.03) compared to those with ischemic heart disease. A trend towards lower risk of CV death was observed in patients with hypertensive heart disease (adjusted HR 0.78, 95% CI 0.61–1.00, *p* = 0.05). Sensitivity analysis in patients with baseline NT-proBNP > 1000 ng/L and in patients with NYHA III-IV symptoms are shown in Appendix A, respectively. In the former group, patients with DCM, ACM, and DICM had a lower adjusted risk of CV mortality, whereas in patients with NYHA III-IV symptoms, no significant differences in survival among the HF etiologies were found.

### 3.3. HF Etiology and Mode of Death

Figure 2 shows the causes of death according to HF etiology. CV death was the most common cause of death in patients with ischemic heart disease (59.8% of deaths) and hypertrophic CM (60.0% of deaths) but lowest in patients with drug-induced CM (25.0% of deaths). Among CV deaths, progressive HF was the most common cause of death in the overall cohort (30.6% of total deaths), ranging from 17.1% in patients with alcoholic CM to 50.0% of total deaths in patients with hypertrophic CM. Sudden death represented 14.4% of total deaths in the overall cohort and was the most common cause of death in patients with dilated CM (17.6% of deaths). Further details on non-CV causes of death are shown in Appendix A.

## 4. Discussion

The main findings of our study were that patients with dilated CM had a substantially better prognosis, whereas patients with drug-induced CM had significantly lower survival than those with other causes of HF. In addition, most of the etiologies we considered were associated with a lower risk of CV death compared with ischemic heart disease.

Over the last three decades, the prognosis of patients with HF has improved remarkably. Angiotensin-converting enzyme inhibitors (ACEIs), beta-blockers, mineralocorticoid receptor antagonists (MRAs), angiotensin receptor-neprilysin inhibitors (ARNIs) [1,2,3], and sodium-glucose co-transporter-2 inhibitors (SGLT-2i) [15,16,17] have significantly reduced the incidence of CV death, particularly from progressive HF and sudden death, with no impact on death from non-CV causes. In addition to medical treatment, a large body of evidence shows that a variety of clinical factors (e.g., age, low LVEF, NYHA class, low systolic blood pressure, chronic kidney disease, and diabetes) may significantly influence the survival of patients with HF [4,18,19]. Even the underlying cause of HF may have a significant impact on the patients’ prognosis. In particular, patients with ischemic heart disease have been shown to have higher mortality than those with non-ischemic causes of HF [5,6]. Felker et al. [8] analyzed more specific causes of HF and found that therapy with doxorubicin, infiltrative CM, and HIV infection carry a significantly worse prognosis than other etiologies. In a more recent study, Pecini et al. [9] showed that patients with valvular heart disease, ischemic heart disease, or dilated CM have the lowest survival, with 1-year mortality rates up to 30%. Conversely, we found that dilated CM is associated with a significantly lower risk of all-cause death compared with ischemic heart disease. This difference could be due to the fact that our study was more recent and patients were treated with better HF therapies. In our cohort, prescription rates for neurohormonal antagonists in patients with dilated CM were 95.4% for ACEIs, angiotensin II receptor blockers, or ARNIs, 91.6% for beta-blockers, and 76.1% for MRAs. We also found that patients with drug-induced CM carried the worst prognosis. This finding is in line with previous evidence that patients with drug-induced CM have a 3- to 4-fold risk of death compared to patients with other etiologies of HF [9,20]. HF and myocardial dysfunction are two of the most common complications of cancer therapies and associated with increased morbidity and mortality [21]. Managing cardiotoxicity can be challenging because the onset of HF may compromise cancer care. In our cohort, patients with drug-induced CM had a short HF duration (median 2 months), which may reflect the onset of cardiotoxicity during or right after chemotherapy. Therefore, the diagnosis of HF may have led to less aggressive chemotherapy regimens and a higher mortality rate in this subgroup of patients. Furthermore, although patients with drug-induced CM died mostly of non-CV causes (which represented 75% of total deaths), in particular of malignancy (59.4% of total deaths), and had a lower risk of CV death compared to patients with ischemic heart disease, almost one-fourth of the deaths in this group were due to progressive HF. Therefore, we speculate that the optimization of HF treatment in these patients may allow the continuation of cancer treatments at high doses by improving HF symptoms and may eventually prevent death from HF. With regards to alcoholic CM, alcohol abuse is known to be a strong risk factor for mortality and morbidity [22]. A study that included more than 300,000 participants showed that heavy drinking is associated with all-cause and cancer-specific mortality but not with CV mortality [23]. Similarly, we found that patients with alcoholic CM had a lower risk of CV death than those with ischemic heart disease, a finding that may be due to the younger age and lower CV risk profile of the former group. Nevertheless, we found no significant differences in terms of all-cause death between the two groups, which could be related to competing risks other than CV disease having a strong impact on the prognosis of patients with alcoholic CM. In this group, almost two-thirds of all deaths were due to non-CV causes. Unfortunately, we do not have data on actual adherence to drinking cessation.

Although historically associated with poor prognosis, hypertrophic CM is now a treatable disease with mortality reduced to as low as 0.5% per year, mainly due to the use of implantable cardioverter-defibrillators [24]. The prevention of sudden cardiac death has resulted in a shift to HF as the major cause of death in patients with hypertrophic CM. Furthermore, data from a contemporary population with hypertrophic CM showed that deaths from non-CV causes exceeded CV causes, especially in patients aged ≥ 60 years [25]. In our cohort, hypertrophic CM was associated with a lower risk of CV death than ischemic heart disease in both the unadjusted and adjusted analyses. However, this did not result in significant differences between the two groups in regards to overall survival. This finding may be due to the higher incidence of non-CV death in patients with hypertrophic CM than those with ischemic heart disease. Moreover, in line with previous evidence, progressive HF represented the main cause of death in patients with hypertrophic CM [24]. Closer follow-up, medical treatment optimization, and prompt identification of patients requiring heart transplantation could be important strategies for preventing these deaths. Our study has several limitations. First, the study cohort was a general HF population treated at a specific multidisciplinary HF clinic in a tertiary care hospital at which most patients were referred from the Cardiology Department. Thus, there was a predominance of relatively young men with HF of ischemic etiology and depressed LVEF, and the population was almost exclusively white. Therefore, we may not be able to fully extrapolate the results to other populations. Notably, a common treatment protocol was applied to all patients, limiting the possible bias introduced by different management strategies or treatment protocols. Furthermore, the relative low number of deaths from alcoholic, chemotherapy-induced, and hypertrophic CM may have led to statistical bias that could affect the results. As already mentioned, we do not have data on actual adherence to drinking cessation, which could have influenced prognosis, though this issue is controversial [26]. Finally, an important limitation, inherent to the observational design of the study, is the great heterogeneity in the baseline patient characteristics among the different etiologies. Although we tried to overcome some of such differences performing the multivariable analysis, we cannot discard some bias in the results.

## 5. Conclusions

In a cohort of real-life ambulatory patients with HF of different etiologies from a specialized HF clinic in a tertiary center, dilated CM and drug-induced CM had the lowest and highest risk of all-cause death, respectively. Among the studied HF etiologies, patients with ischemic heart disease had the highest adjusted risk of CV death compared to those with other etiologies of HF.

## Figures and Tables

**Figure 1 jcm-11-00784-f001:**
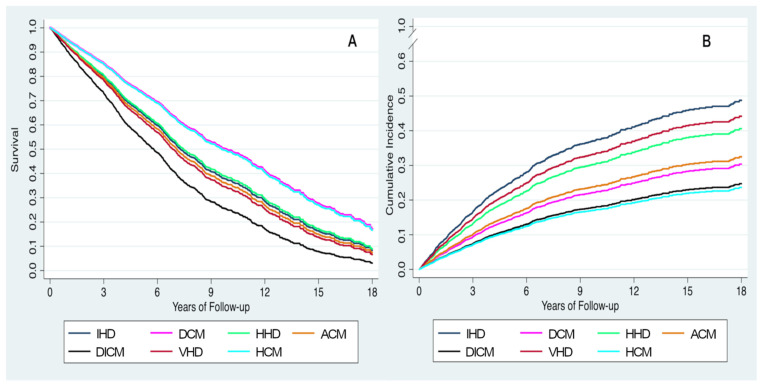
Adjusted survival curves (**A**) and adjusted incidence curves for cardiovascular death (**B**) based on heart failure etiology. ACM, alcoholic cardiomyopathy; DCM, dilated cardiomyopathy; DICM, drug-induced cardiomyopathy; HCM, hypertrophic cardiomyopathy HHD, hypertensive heart disease; IHD: ischemic heart disease; VHD, valvular heart disease.

**Figure 2 jcm-11-00784-f002:**
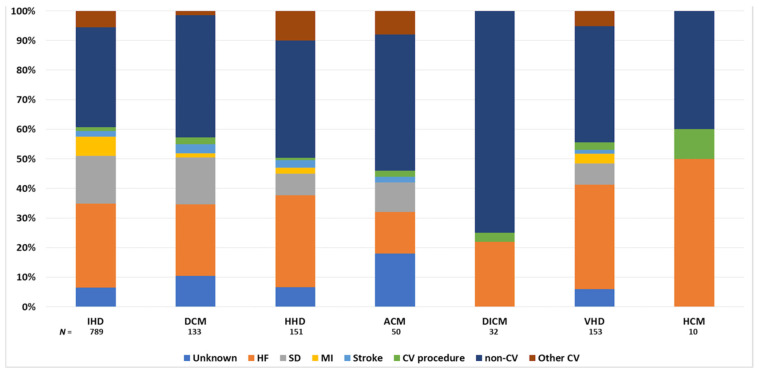
Causes of death according to the etiology of heart failure. ACM, alcoholic cardiomyopathy; CV, cardiovascular; DCM, dilated cardiomyopathy; DICM, drug-induced cardiomyopathy; HCM, hypertrophic cardiomyopathy; HF, heart failure; HHD, hypertensive heart disease; IHD: ischemic heart disease; MI, myocardial infarction; SD, sudden death; VHD, valvular heart disease.

**Table 1 jcm-11-00784-t001:** Baseline characteristics by etiology of heart failure.

	Total Cohort (*n* = 2387)	IHD (*n* = 1222)	DCM (*n* = 415)	HHD (*n* = 231)	ACM (*n* = 120)	DICM (*n* = 73)	VHD (*n* = 237)	HCM (*n* = 89)	*p*-Value
Age, years	67 ± 13	68 ± 11	62 ± 14	71 ± 13	57 ± 10	59 ± 12	71 ± 11	58 ± 15	<0.001
Male	1702 (71.3)	969 (79.3)	300 (72.3)	119 (51.5)	113 (94.2)	18 (24.7)	117 (49.4)	66 (74.2)	<0.001
White	2309 (96.7)	1187 (97.1)	392 (94.5)	226 (97.8)	117 (97.5)	71 (97.3)	23 3(98.3)	83 (93.3)	0.03
HF duration, months	8 (2–48)	7 (1–42)	6 (2–36)	11 (2–48)	5 (2–36)	2 (1–14)	24 (4–42)	24 (4–118)	<0.001
NYHA class III/IV	659 (27.6)	345 (28.2)	82 (19.8)	84 (36.4)	15 (12.5)	21 (28.8)	99 (41.8)	13 (14.6)	<0.001
LVEF, %	35.4 ± 14.2	32.59 ± 10.4	30.2 ± 10.7	44.4 ± 17.0	26.7 ± 10.1	33.9 ± 12.6	44.6 ± 16.0	64.9 ± 11.2	<0.001
Diabetes	1040 (43.6)	652 (53.4)	140 (33.7)	105 (45.5)	26 (21.7)	19 (26.0)	83 (35.0)	15 (16.9)	<0.001
Hypertension	1533 (64.2)	821 (67.2)	215 (51.8)	225 (97.4)	51 (42.5)	25 (34.2)	156 (65.8)	40 (44.9)	<0.001
COPD	408 (17.1)	218 (17.8)	70 (16.9)	41 (17.7)	30 (25.0)	2 (2.7)	43 (18.1)	4 (4.5)	<0.001
PVD	356 (14.9)	255 (20.9)	39 (9.4)	29 (12.6)	11 (9.2)	1 (1.4)	19 (8.0)	2 (2.2)	<0.001
Anaemia ^1^	1072 (45.0)	657 (53.9)	104 (25.1)	113 (48.9)	30 (25.0)	25 (34.2)	129 (54.4)	14 (15.9)	<0.001
Renal insufficiency ^2^	1051 (44.0)	587 (48.0)	138 (33.3)	144 (62.3)	21 (17.5)	26 (35.6)	121 (51.1)	14 (15.7)	<0.001
Atrial fibrillation or flutter	518 (21.7)	173 (14.2)	83 (20.0)	82 (35.5)	28 (23.3)	6 (8.2)	130 (54.9)	16 (18.0)	<0.001
BMI, kg/m^2^ (*n* = 2350)	27 (24–30)	27 (24–30)	27 (24–30)	29 (36–34)	26 (23–30)	26 (23–30)	26 (23–30)	27 (25–30)	<0.001
NT-proBNP, ng/L (*n* = 1712)	1703 (730–4061)	2193 (967–4978)	1118 (517–2470)	1772 (785–4117)	1075 (502–2799)	1409 (544–3491)	2129 (912–4460)	863 (307–1739)	<0.001
Treatment (during follow-up)									
ACEI/ARB/ARNI	2052 (86.0)	1070 (87.6)	393 (94.7)	180 (77.9)	116 (96.7)	69 (94.5)	182 (76.8)	42 (47.2)	<0.001
Beta-blocker	2137 (89.5)	1136 (93.0)	380 (91.6)	184 (79.7)	113 (94.2)	68 (93.2)	182 (76.8)	74 (83.1)	<0.001
MRA	1581 (66.2)	808 (66.1)	316 (76.1)	133 (57.6)	92 (76.7)	49 (67.1)	164 (69.2)	19 (21.3)	<0.001
Loop diuretic	2154 (90.2)	1122 (91.8)	365 (88.0)	224 (97.0)	116 (96.7)	58 (79.5)	227 (95.8)	42 (47.2)	<0.001
Digoxin	889 (37.2)	398 (32.6)	162 (39.0)	96 (41.6)	68 (56.7)	22 (30.1)	137 (57.8)	6 (6.7)	<0.001
Ivabradine	524 (22.0)	271 (22.2)	130 (31.3)	32 (13.9)	35 (29.2)	32 (43.8)	21 (8.9)	3 (3.4)	<0.001
CRT	267 (11.2)	126 (10.3)	79 (19.0)	17 (7.4)	11 (9.2)	6 (8.2)	26 (11.0)	2 (2.2)	<0.001
ICD	369 (15.5)	237 (19.4)	71 (17.1)	9 (3.9)	14 (11.7)	4 (5.5)	19 (8.0)	15 (16.9)	<0.001

Data are given as mean ± standard deviation, median (interquartile range), or *n* (%). ACEI, angiotensin-converting enzyme inhibitor; ACM, alcoholic cardiomyopathy; ARB, angiotensin II receptor blocker; ARNI, angiotensin receptor–neprilysin inhibitor; BMI, body mass index; COPD, chronic obstructive pulmonary disease; CRT, cardiac resynchronization therapy; DCM, dilated cardiomyopathy; DICM, drug-induced cardiomyopathy; HF, heart failure; HCM, hypertrophic cardiomyopathy; HHD, hypertensive heart disease; ICD, implantable cardioverter-defibrillator; IHD: ischemic heart disease; LVEF, left ventricular ejection fraction; MRA, mineralocorticoid receptor antagonist; NYHA, New York Heart Association; NT-proBNP, N-terminal pro-brain natriuretic peptide; PVD, peripheral vascular disease; VHD, valvular heart disease. ^1^ According to World Health Organization criteria (<13 g/dL in men and <12 g/dL in women). ^2^ Estimated glomerular filtration rate (Chronic Kidney Disease-Epidemiology Collaboration equation) <60 mL/min/1.73 m^2.^.

**Table 2 jcm-11-00784-t002:** Cox regression multivariable analysis for all-cause death and for cardiovascular mortality.

	Unadjusted Analysis						Adjusted Analysis *					
	All-Cause Death			Cardiovascular Death ^†^			All-Cause Death			Cardiovascular Death ^†^		
	HR	95%CI	*p*-Value	HR	95%CI	*p*-Value	HR	95%CI	*p*-value	HR	95%CI	*p*-Value
IHD	1			1			1			1		
DCM	0.48	0.40–0.57	<0.001	0.40	0.31–0.53	<0.001	0.68	0.56–0.83	<0.001	0.54	0.42–0.70	<0.001
HHD	1.03	0.87–1.23	0.72	0.86	0.68–1–08	0.21	0.96	0.80–1.16	0.68	0.78	0.61–1.00	0.05
ACM	0.49	0.37–0.65	<0.001	0.33	0.21–0.52	<0.001	0.93	0.67–1.28	0.65	0.59	0.37–0.94	0.03
DICM	0.73	0.51–1.04	0.08	0.27	0.14–0.52	<0.001	1.47	1.02–2.11	0.04	0.42	0.22–0.83	0.01
VHD	1.20	1.01–1.42	0.04	0.97	0.76–1.23	0.79	1.08	0.89–1.30	0.44	0.87	0.68–1.12	0.29
HCM	0.27	0.14–0.50	<0.001	0.26	0.12–0.57	<0.001	0.75	0.40–1.41	0.37	0.40	0.18–0.90	0.03

* Adjusted for heart failure duration, New York Heart Association class III-IV, diabetes, anemia, renal insufficiency, chronic obstructive pulmonary disease, and peripheral arteriopathy. † Using Fine and Gray competing risk. ACM, alcoholic cardiomyopathy; CI: confidence interval; DCM, dilated cardiomyopathy; DICM, drug-induced cardiomyopathy; HCM, hypertrophic cardiomyopathy; HHD, hypertensive heart disease; HR: hazard ratio; IHD: ischemic heart disease; VHD, valvular heart disease.

## Data Availability

Data collected for the study and related documents will not be made available to others.

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
