# Peer review of "Cause of Death in Heart Failure Based on Etiology: Long-Term Cohort Study of All-Cause and Cardiovascular Mortality"

_jcm, 2022, doi:10.3390/jcm11030784_

Round 1
Reviewer 1 Report
Dear colleagues,
congratulation to your article. I find the result of your study very intersting, important and usefull. For example, interesting finding is the difference in the baseline medical therapy in different aetiologies of heart failure.
I have some questions to the authors:
- Can you comment on the difference of the proportion of patients receiving ivabradine in patients with dilated cardiomyopathy compared to ischemic cardiomyopathy?
- Can you comment on the proportion of digoxin use (56.7 %) in patients with alcoholic cardiomyopathy provided that atrial fibrillation was present in about 23 % of ACM subjects?
Author Response
Reviewer 1
Comment 1. Can you comment on the difference of the proportion of patients receiving ivabradine in patients with dilated cardiomyopathy compared to ischemic cardiomyopathy?
Reply to Comment 1. We appreciate the Reviewer’s comment. We think that this may be an incidental finding since all patients admitted to our clinic were managed according to the ESC heart failure (HF) guidelines and the etiology of HF was not considered for starting ivabradine. In addition, we may speculate that patients with dilated cardiomyopathy could have a higher prevalence of hypotension as compared with those with ischemic cardiomyopathy and, therefore, in the former group ivabradine initiation was preferred over beta blockers up-titration.
Comment 2. Can you comment on the proportion of digoxin use (56.7 %) in patients with alcoholic cardiomyopathy provided that atrial fibrillation was present in about 23 % of ACM subjects?
Reply to Comment 2. We thank the Reviewer for the comment. In our cohort, patients with alcoholic cardiomyopathy had a significantly lower ejection fraction compared with other etiologies (26.7% vs. 35.8%, respectively, p<0.001). Therefore, the higher use of digoxin might be related to the attempt to optimize the HF treatment in these patients with very low ejection fraction regardless of the presence of atrial fibrillation, given that, according to ESC guidelines, “digoxin may be considered in patients with symptomatic HFrEF in sinus rhythm despite treatment with an ACE-I (or ARNI), a beta- blocker and an MRA, to reduce the risk of hospitalization”.

Reviewer 2 Report
Dear Authors,
Thank you for allowing me to revise the original manuscript entitled: Cause of death in heart failure based on etiology: Long-term cohort study of all-cause and cardiovascular mortality.
The manuscript is exciting but has, in my opinion, one major limitation that may have flawed all the data analysis.
The baseline features appear extremely heterogeneous since the absence of a clear definition of the referral reasons to the heart failure center. The patients are not categorized as advanced heart failure nor with stage A, B, C, D classes either not as Weber classes, NYHA or INTERMACS. Indeed, it is expected that, as many heart failure centers, they tend to concentrate the illest patients affected by IHD and identify and collect from the early diagnosis the Hypertrophic Cardiomyopathy patients. I suggest the authors deal with this significant limitation for the sake of achieving a more excellent scientific sound. Some of the differences in the results between this and other reports appear flawed from the heterogeneity of clinical conditions with no clinical factors itemizing the analysis.
Looking at the data, in fact, Ejection fraction, Natriuretic Factors, but also HF duration, age, and many other variables commonly associated with different survivals are significantly different. The authors should attempt to manage the effects of this heterogeneity by selecting a category of patients to be examined (i.e., EF< 35%, NYHA >3, Advanced HF, More than two hospitalizations during last year, INTERMACS or Weber Classes, Pro-BNP> 1000).
Only standardizing the entry point the outcome may fairly be investigated assessing the effect of the etiology over other risk factors.
Minor comments: Figure 2 is of significant interest and may be analyzed as competing risk or with a more dynamic analysis assessing the clinical trajectory of the patients with different etiology.
The number of unknown causes of death is not acceptable. The authors should attempt to clarify both the unknown reasons and the non-CV. It could be of interest to distinguish neoplastic from other non-CV causes since the relevance for the patients with DICM.
My best regards
Author Response
Reviewer 2
Comment 1. The baseline features appear extremely heterogeneous since the absence of a clear definition of the referral reasons to the heart failure center. The patients are not categorized as advanced heart failure nor with stage A, B, C, D classes either not as Weber classes, NYHA or INTERMACS. Indeed, it is expected that, as many heart failure centers, they tend to concentrate the illest patients affected by IHD and identify and collect from the early diagnosis the Hypertrophic Cardiomyopathy patients. I suggest the authors deal with this significant limitation for the sake of achieving a more excellent scientific sound. Some of the differences in the results between this and other reports appear flawed from the heterogeneity of clinical conditions with no clinical factors itemizing the analysis. Looking at the data, in fact, Ejection fraction, Natriuretic Factors, but also HF duration, age, and many other variables commonly associated with different survivals are significantly different. The authors should attempt to manage the effects of this heterogeneity by selecting a category of patients to be examined (i.e., EF< 35%, NYHA >3, Advanced HF, More than two hospitalizations during last year, INTERMACS or Weber Classes, Pro-BNP> 1000).
Only standardizing the entry point the outcome may fairly be investigated assessing the effect of the etiology over other risk factors.
Reply to Comment 1. We agree with the Reviewer. The criteria for referral are quite comprehensive and are reported in the manuscript and many previous published manuscripts (see references 10 and 11 of the manuscript). It is true that baseline features are very heterogeneous but we think that this is an inherent limitation when reporting data of real-life patients with HF from different etiologies like those that were attended at our HF Unit. Nevertheless, in our study, we tried to solve some of these differences by analyzing the role of etiology on mortality with the multivariable analysis. According to the reviewer suggestion, in the new version of the manuscript, we addressed this issue in the limitations section: “Finally, an important limitation, inherent to the observational design of the study, is the great heterogeneity in the baseline patient characteristics among the different etiologies. Although we tried to overcome some of such differences performing the multivariable analysis, we cannot discard some bias in the results.” (Page 11, line 313-316).
Moreover, according to the reviewer’s suggestion, we performed a sensitivity analysis including patients in NYHA class III-IV and patients with NT-proBNP levels > 1000 ng/L. We provided a supplementary table with such data in the revised version of the manuscript (Table S2 and Table S3, Supplementary Material) and modified the manuscript as follows: “Sensitivity analysis in patients with baseline NT-proBNP > 1000 ng/L and in patients with NYHA III-IV symptoms are shown in Table S1 and Table S2, respectively. In the former group, patients with DCM, ACM and DICM had a lower adjusted risk of CV mortality, whereas in patients with NYHA III-IV symptoms, no significant differences in survival among the HF etiologies were found” (page 7, line 200-204)
Comment 2. Figure 2 is of significant interest and may be analyzed as competing risk or with a more dynamic analysis assessing the clinical trajectory of the patients with different etiology.
Reply to Comment 2. We appreciate the Reviewer’s comment. Since in Figure 1 we already depicted the results from the survival analysis using competing risk methodology over time, our aim in Figure 2 was to describe simply the specific causes of death across the different etiologies. Therefore, if the Reviewer agrees, we kindly ask to keep Figure 2 as it is.
Comment 3. The number of unknown causes of death is not acceptable. The authors should attempt to clarify both the unknown reasons and the non-CV. It could be of interest to distinguish neoplastic from other non-CV causes since the relevance for the patients with DICM.
Reply to Comment 3. We thank the Reviewer for the comment. Unfortunately, we are not able to provide further details on the deaths for unknown causes. Nevertheless, in our cohort, they represented the 7.1% of the total deaths (93/1317). This finding is in line with previous evidence. In an analysis from the Framingham Heart Study, the rate of death from unknown cause was 9%. In addition, a more recent sub analysis from the PARADIGM-HF showed that unknown represented almost 5% of total deaths.
As suggested by the Reviewer, we provided a supplementary table with further details on non-CV deaths and the rates of death due to malignancy (Table S3, Supplementary Material). We modified the manuscript as follows: “Furthermore, although patients with drug-induced CM died mostly of non-CV causes (which represented 75% of total deaths), in particular of malignancy (59.4% of total deaths), and had a lower risk of CV death compared to patients with ischemic heart disease, almost one-fourth of the deaths in this group were due to progressive HF” (page 10, line 274-278).

Round 2
Reviewer 2 Report
Dear Authors,
thank you for your careful answer to the reviewer's concerns and for the minimal but appropriate revisions to the manuscript.
Best regards